# Individual differences in motives for costly punishment
Scott Claessens [1,2] ✉, Quentin D. Atkinson [2] & Nichola J. Raihani [2,3] ✉

Costly punishment is thought to be a key mechanism sustaining human cooperation. However, the motives for punitive behaviour remain unclear. Although often assumed to be motivated by a desire to convert cheats into cooperators, punishment is also consistent with other functions, such as levelling payoffs or improving one's relative position. We used six economic games to tease apart different motives for punishment and to explore whether different punishment strategies were associated with personality variables, political ideology, and religiosity. We used representative samples from the United Kingdom and the United States ($N = 2010$) to estimate the frequency of different punishment strategies in the population. The most common strategy was to never punish. For people who did punish, strategy use was more consistent with egalitarian motives than behaviour-change motives. Nevertheless, different punishment strategies were also associated with personality, social preferences, political ideology, and religiosity. Self-reports of behaviour in the games suggested that people have some insight into their punishment strategy. These findings highlight the multipurpose nature of human punishment and show how the different motives underpinning punishment decisions are linked with core character traits.

Humans cooperate on a scale that is unparalleled in the animal kingdom. One mechanism thought to sustain this level of cooperation is costly punishment, whereby individuals harm others at a personal cost[1], ostensibly encouraging cooperative behaviour from the target or bystanders[2–4]. Punishment offers a route to maintaining or increasing cooperation by changing the payoff structure of social interactions such that it no longer pays to cheat or exploit social partners[1,5]. Yet, despite its theoretical importance, the question of why people choose to punish others is still hotly contested[6]. In this study, we use a battery of economic games to disentangle the different motives underpinning punishment and explore how these motives vary across individuals.

In humans, punishment is often studied in laboratory settings using economic games[6–15]. In some of these games, participants are given a sum of money that they can use to invest in collective action or to help others; conversely, they can 'cheat' by keeping the money for themselves or by exploiting the contributions of others. In other games, participants are endowed with sums of money and can then 'cheat' by destroying a partner's payoffs or stealing from them[16,17]. Punishment is introduced into such games by giving participants the option to pay a small 'fee' to impose a greater 'fine' on their co-players. Several lines of experimental evidence indicate that people use this punishment option[13], that they enjoy punishing[18], and that they frequently, though not always[19], punish those who cheat, free-ride, or steal[11,12].

Evidence from these experiments suggests that the threat of costly punishment plays an important role in promoting human cooperation. People tend to cooperate more in games where punishment is possible compared to those where it is not[6–8]. Similarly, people make higher offers in the Ultimatum Game (where punishment is possible) compared to the structurally-similar Dictator Game (where it is not)[20]. This cooperation-enhancing effect of punishment has also been observed across societies[8], leading some to suggest that costly punishment has played a key role in the cultural evolution of cooperation in humans[21–24].

Nevertheless, it remains unclear whether individuals playing economic games use punishment as a behaviour-change tool to enforce cooperation or to achieve other ends. Some have argued that punishment is primarily used to shape behaviour, either to deter personal harm[3,10,25] or to uphold normative standards of cooperative behaviour[22,23,26–29]. But while the *threat* of punishment can have a cooperation-enhancing effect, the *enactment* of this punishment does not consistently deter targets from cheating in the future[6]. This calls into question whether punishment primarily operates as a behaviour-change tool or whether it is used to achieve other goals.

Beyond behaviour shaping concerns, there are other reasons why people punish in economic games. Punishers might be motivated by a desire for revenge rather than deterrence, punishing in response to harm that was personally incurred[30]. Punishment might be driven by concerns about relative payoffs, such as disadvantageous inequity aversion (i.e., avoiding

[1]School of Psychology, University of Kent, Canterbury, UK. [2]School of Psychology, University of Auckland, Auckland, New Zealand. [3]Department of Experimental Psychology, University College London, London, UK. ✉e-mail: scott.claessens@gmail.com; n.raihani@ucl.ac.uk

having less than others)[6,17] and/or general egalitarian preferences (i.e., wanting all participants to receive the same payoffs)[31]. Such concerns about relative payoffs may be activated when participants earn less than cheats in economic games or when there are income disparities in these settings. People might also use punishment for competitive purposes, seeking advantageous inequity for themselves (i.e., having more than others) and/or improving their relative position[6].

Common economic game designs have been unable to tease apart these different motives for punishment because participants who interact with cheats in these games experience both losses and lower relative payoffs. The typical 1:3 fee-fine ratio of punishment in economic games compounds this issue. With this setup, people can simultaneously use punishment to reciprocate losses, to deter others from cheating, and to reduce or reverse disparities in payoffs between themselves and targets. To add to this complexity, it is evident that people use punishment in seemingly disparate ways: punishing when no behaviour change is possible, such as in one-shot games[13,17,32,33], on the very last round of repeated games[34], or in games where the target never learns about the punishment[35]; punishing those who did not cheat or who over-contributed to collective action (sometimes called 'antisocial punishment')[19,36]; punishing in scenarios where they were not personally harmed (third-party punishment)[37]; and punishing in scenarios where disparities in payoffs did not arise from participants' actions[31,38,39].

The general conclusion from this research is that there is no one unifying function of costly punishment in humans. Instead, punishment should be thought of as a flexible behavioural tool that serves a variety of functions that are not mutually exclusive[6]. Due to its multipurpose nature, we should therefore expect variation in punishment strategies in the population, much like the observed variation in social learning strategies[40]. Some individuals may use punishment as a behaviour shaping tool, for example, while others may use it to reduce or reverse payoff differentials. This raises the underexplored question of which punishment strategies are more frequent in human populations.

Here, we aim to delineate nine possible punishment strategies by asking whether people punish in a manner consistent with a specific strategy across a suite of economic games. Table 1 summarises the potential strategies for costly punishment in the economic games that we considered, and the behavioural patterns they predict. These proximate strategies are consistent with different ultimate functions of punishment that have been emphasised in the literature, such as changing how others behave towards the punisher[1,15], changing relative payoffs between the punisher and the target[6,41], and improving cooperation within the cultural group such that the group is more successful than other groups[22]. Note that Table 1 is not an exhaustive list of all possible punishment strategies: we do not include punishment strategies that serve reputational functions, such as signalling trustworthiness[4,42–45], because our focus is on punishment strategies in anonymous economic games without reputational incentives (but see ref. 46).

Building on previous designs that have used one-shot economic games to explore behaviour-shaping and egalitarian motives for punishment[17,32,35,47,48], we employ a suite of one-shot economic games where individuals are given the opportunity to punish targets at a personal cost (Fig. 1). In each game, targets either steal from another individual or do nothing. Representative samples of participants from the United Kingdom ($n = 1014$) and the United States ($n = 996$) completed all six games online. We designed the suite of games to tease apart the proposed punishment strategies in Table 1, such that each strategy predicts a different pattern of behaviour across all the games (see Methods for more detail about the six games). We use the resulting behavioural patterns to discern which punishment strategy participants are employing.

We also explore whether individual differences in personality, religiosity, political ideology, and demographics are associated with variation in punishment strategies. Prior work has shown that punitive behaviour in economic games is predicted by personality, social preferences, religiosity,

and political ideology. For example, people who are more agreeable or more prosocial are less likely to punish in economic games[49–51], whereas political conservatives and those higher on measures of right-wing authoritarianism express more punitive sentiment[52,53]. Other associations are less clear: for example, religiosity can show a positive, negative, or no association with punitive behaviour[54–56]. However, because previous studies have not distinguished between different punishment strategies, the proximate motivations underlying these trait correlations remain unclear. We aim to fill this gap by linking our economic game data to a host of individual difference measures collected in a separate survey. Although we pre-registered our study design and analysis plan (see 'Methods'), we note that we did not pre-register any directional hypotheses for these exploratory analyses.

Finally, we ask whether people have insight into their own punishment strategy. Prior work has argued that people are often unaware of the underlying function of their punitive behaviour, yet they feel compelled to enact it anyway[30]. We assess this claim by asking people to explicitly report the reasons for their behaviour in the games, exploring whether these self-reported reasons align with patterns of punitive behaviour.

## Methods
### Pre-registration
We pre-registered the study on the Open Science Framework before collecting data in the United Kingdom (11th November 2022; https://osf.io/k75fc). We submitted another pre-registration before collecting data in the United States (20th June 2023; https://osf.io/q4hdy). In the pre-registrations, we outlined our study design, exclusion criteria, and analysis plan. As the study was exploratory, we did not pre-register any explicit hypotheses. We did not deviate from the pre-registrations.

### Exclusion criteria
We pre-registered that we would exclude participants who failed any of the attention checks, sped through the surveys (i.e., two standard deviations below the median duration), or flatlined (i.e., provided identical responses to matrix questions). We also stated that we would exclude data for particular games if participants failed the comprehension question for that game. We followed our pre-registered plan of conducting analyses with and without these exclusions (analyses without exclusions are reported in the Supplementary Material).

### Participants
We collected a representative sample of 1019 participants from the United Kingdom through the online platform Prolific (https://www.prolific.com/). All of these participants completed the economic games, and 973 returned to complete the follow-up survey a week later (95% retention rate). After exclusions, we were left with 1014 participants overall (513 female, 481 male, 20 unspecified gender; 72 Asian, 30 Black, 16 Mixed, 865 white, 11 Other, 20 unspecified ethnicity; mean age = 45.78; see Supplementary Fig. 1).

We later collected a representative sample of 1005 participants from the United States through Prolific. All of these participants completed the economic games and 957 returned to complete the follow-up survey (95% retention rate). After exclusions, we were left with 996 participants overall (504 female, 482 male, 10 unspecified gender; 53 Asian, 127 Black, 19 Mixed, 774 white, 13 Other, 10 unspecified ethnicity; mean age = 45.65; see Supplementary Fig. 2).

### Materials
**Economic games.** In the first part of the study, participants completed six economic games, each with slight variations. In all games, the participant takes the role of P1. P2 either (a) steals £0.20 from another player and adds it to their payoff or (b) does nothing. Following the strategy method, participants responded to each of these possible decisions. For each of these cases, participants are asked whether they would like to pay money to reduce P2's payoff. Games A-E have two players, and Game F has three players.

**Table 1 | Summary of the different functions for punishment and the behavioural strategies they predict**

| Strategy | Behavioural description | Game A (AI) 70–10 | | Game B (Equal) 70–30 | | Game C (Computer) 70–30 | | Game D (1:1 Fee Fine) 70–30 | | Game E (DI) 70–50 | | Game F (Third Party) 70–70 [100] | |
|---|---|---|---|---|---|---|---|---|---|---|---|---|---|
| | | Steal 50–30 | No steal 70–10 | Steal 50–50 | No steal 70–30 | Steal 50–50 | No steal 70–30 | Steal 50–50 | No steal 70–30 | Steal 50–70 | No steal 70–50 | Steal 50–90 [100] | No steal 70–70 [100] |
| Deterrent | Punish to deter another who has harmed you from harming you again in the future | ✓ | × | ✓ | × | × | × | ✓ | × | ✓ | × | × | × |
| Norm-enforcing | Punish to enforce a shared anti-harm norm and encourage future norm compliance, even amongst third parties | ✓ | × | ✓ | × | × | × | ✓ | × | ✓ | × | ✓ | × |
| Revenge | Punish if doing so harms another who has harmed you | ✓ | × | ✓ | × | ✓ | × | ✓ | × | ✓ | × | × | × |
| Avoid DI | Punish if doing so avoids disadvantageous inequity for self | × | × | × | × | × | × | × | × | ✓ | × | × | × |
| Egalitarian | Punish if doing so makes payoffs for all more equal | × | × | × | × | × | × | × | × | ✓ | × | ✓ | × |
| Seek AI | Punish if doing so produces advantageous inequity for self | × | × | ✓ | × | ✓ | × | × | × | × | × | × | × |
| Competitive | Punish if doing so improves your relative position | ✓ | ✓ | ✓ | ✓ | ✓ | ✓ | × | × | ✓ | ✓ | ✓ | ✓ |
| Antisocial | Punish exclusively those who do not cause harm | × | ✓ | × | ✓ | × | ✓ | × | ✓ | × | ✓ | × | ✓ |
| Never punish | Never punish others | × | × | × | × | × | × | × | × | × | × | × | × |

Games A–F are the games employed in the current study (see 'Methods' for more details). In each of the six games, participants are given the opportunity to punish players who 'steal' and those who do not, meaning that participants make twelve punishment decisions in total. Each behavioural strategy implies a unique pattern of punishment across all decisions (see 'Methods' for detailed explanation of strategies). Ticks reflect decisions to punish, crosses reflect decisions to not punish. In column headers, payoffs at the first stage (above) and the second stage (below) are denoted as P1–P2 (or P2–P3 [P1] for Game F), where participants take the role of P1 and P2 is the target of punishment. AI advantageous inequity, DI disadvantageous inequity.

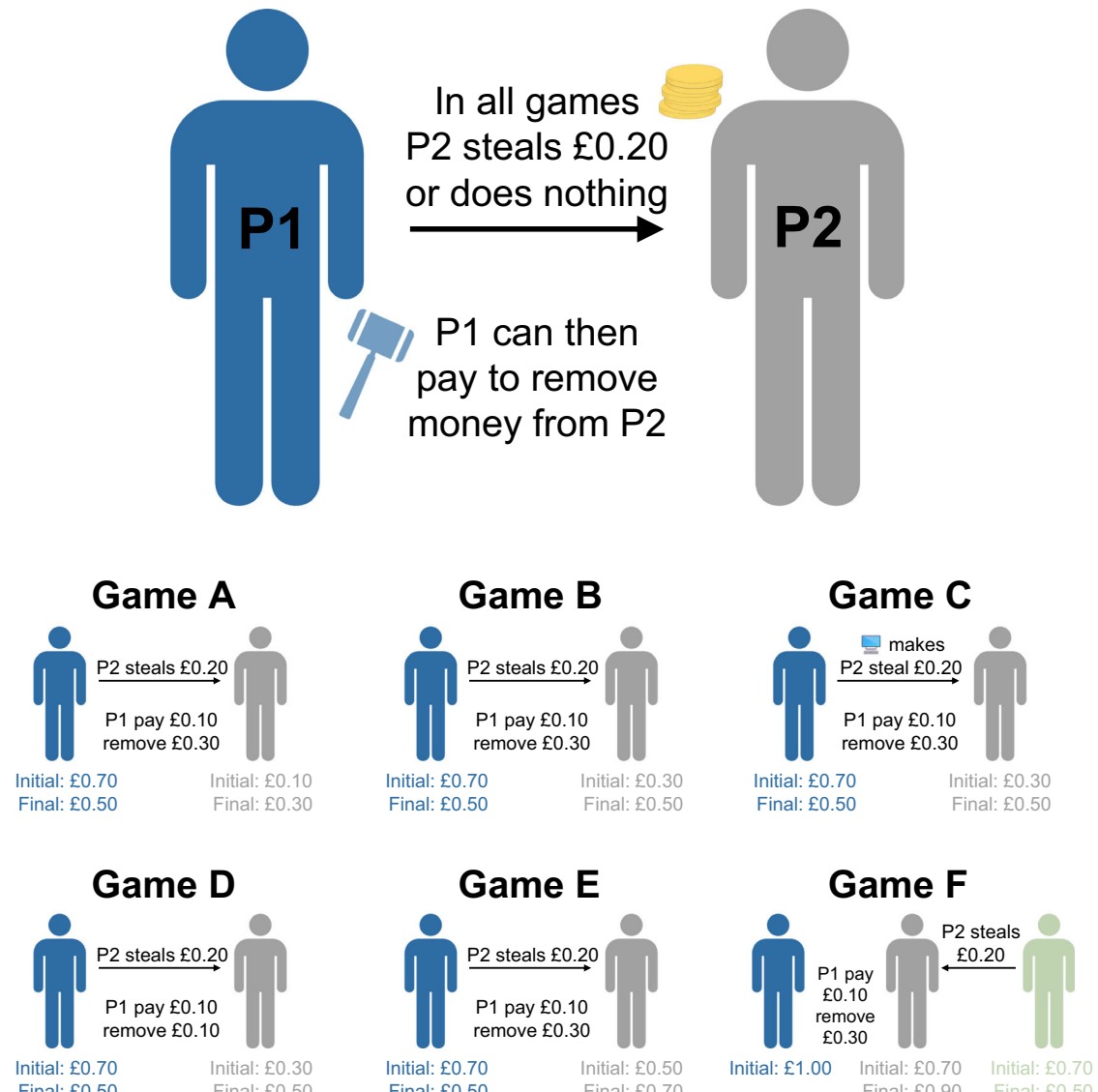

**Fig. 1 | Visual summary of the six economic games.** In the games, Player 2 either steals £0.20 from Player 1 (the focal player) or does nothing. Player 1 is then given the option to punish by paying a certain amount of money to remove money from Player 2 (this money is destroyed). The six games are variants on this general setup, creating situations where (A) Player 2 is still worse off after stealing, (B) Player 2 creates equality by stealing, (C) the computer 'decides' whether Player 2 steals, (D) the fee-fine ratio is 1:1, (E) Player 2 is better off after stealing, and (F) Player 2 steals instead from a third-party.

The six games are as follows (see Fig. 1 for a visual representation of the games):

1. *Game A (Advantageous Inequity).* P1 starts with £0.70 and P2 starts with £0.10. P2 is given the option to either steal £0.20 from P1 or do nothing. P1 can then pay £0.10 to reduce P2's payoff by £0.30.
2. *Game B (Equal).* P1 starts with £0.70 and P2 starts with £0.30. P2 is given the option to either steal £0.20 from P1 or do nothing. P1 can then pay £0.10 to reduce P2's payoff by £0.30.
3. *Game C (Computer).* P1 starts with £0.70, and P2 starts with £0.30. Participants are told that "the computer will decide" whether P2 steals £0.20 from P1 or does nothing. P1 can then pay £0.10 to reduce P2's payoff by £0.30.
4. *Game D (1:1 Fee-Fine).* P1 starts with £0.70 and P2 starts with £0.30. P2 is given the option to either steal £0.20 from P1 or do nothing. P1 can then pay £0.10 to reduce P2's payoff by £0.10.
5. *Game E (Disadvantageous Inequity).* P1 starts with £0.70 and P2 starts with £0.50. P2 is given the option to either steal £0.20 from P1 or do nothing. P1 can then pay £0.10 to reduce P2's payoff by £0.30.

6. *Game F (Third-Party).* P1 starts with £1.00, P2 and P3 start with £0.70. P2 is given the option to either steal £0.20 from P3 or do nothing. P1 can then pay £0.10 to reduce P2's payoff by £0.30.

We delineate nine punishment strategies that can be isolated based on players' decisions across the six economic games (Table 1). In existing literature, it is often unclear whether punishment decisions are sensitive to the previous actions of others or to payoff sensitivity. This is because (i) players' decisions typically introduce payoff differentials in economic games and (ii) where fine > fee, punishment tools can be used to harm co-players and to change relative payoffs[6]. Of the nine strategies we define, *deterrent, norm-enforcing, revenge* and *antisocial* are strategies that are only sensitive to the actions of others. Conversely, *avoid disadvantageous inequity, egalitarian, seek advantageous inequity* and *competitive* are strategies that are sensitive to payoffs. *Never punish* is an unconditional strategy (i.e., no sensitivity to actions or payoffs).

Our strategy definitions sometimes differ from how these strategies are defined and discussed in the literature (e.g., antisocial punishment).

This is essential because previous definitions permit many possible motives, while we are aiming to isolate punishment motives. We define our strategies as follows:

- *Deterrent*: Deter others who have harmed you from harming you again in the future (when the target steals in Games A, B, D, and E).
- *Norm-enforcing*: Punish to enforce a shared anti-harm norm and encourage future norm compliance (when the target steals in Games A, B, D, E and F). Though we cannot measure future norm compliance in our one-shot games, these non-repeated (and 'stranger design') games have been used to identify norm-enforcing motives in many previous studies[11,12].
- *Revenge*: Punish if doing so harms another who harmed you (when the target steals in Games A-E).
- *Avoid disadvantageous inequity*: Punish if doing so avoids disadvantageous inequity for the self (when the target steals in Game E).
- *Egalitarian*: Punish if doing so makes payoffs for all more equal (when the target steals in Games E and F) (c.f. ref. 31).
- *Seek advantageous inequity*: Punish if doing so produces advantageous inequity for the self (when the target steals in Games B and C). This is one possible motive underlying what others have previously called 'antisocial punishment' or 'spite'; here we isolate it.
- *Competitive*: Punish if doing so improves your relative position (all games except Game D).
- *Antisocial*: Punish exclusively those who do not cause harm (punish non-stealing target in all games). This definition differs from some previous definitions of antisocial punishment, but this is essential to isolate the full range of motives that could be driving punishment and which are not typically disentangled in other settings.
- *Never punish*: A person who never incurs a personal cost to punish a co-player (does not punish in any game).

For each game, participants saw the game instructions and answered a comprehension question before providing their decisions. After completing all the games, participants were asked to give an open-ended response explaining their behaviour in the games, and then responded to several slider questions capturing the different reasons for their decisions (for full wordings, see Supplementary Table 1).

## Survey questions

In a follow-up survey, we collected the following data on participants (for wordings of all questions, see Supplementary Table 2):

- *Demographics.* In the survey, we collected information on participants' education level and self-reported socio-economic status (MacArthur ladder[57]). We also collected additional demographic data from Prolific (e.g., age, gender, student status).
- *Personality.* We used the Mini-IPIP scale[58] to measure the Big 6 personality dimensions of agreeableness ($\alpha = 0.83$), conscientiousness ($\alpha = 0.75$), extraversion ($\alpha = 0.83$), honesty-humility ($\alpha = 0.77$), openness to experience ($\alpha = 0.79$), and neuroticism ($\alpha = 0.79$). Four items were used for each personality dimension.
- *Social Value Orientation.* We used the Social Value Orientation Slider Measure to measure other-regarding preferences[59]. Across fifteen items, participants made decisions on how to allocate different amounts of money between themselves and another anonymous individual. From these decisions, we calculated participants' Social Value Orientation 'angle' as a measure of their other-regarding preference, following the steps outlined in ref. 59.
- *Political ideology.* We included several measures of political ideology, including left-right conservatism, Social Dominance Orientation[60] ($\alpha = 0.91$; eight items), and Right Wing Authoritarianism[61] ($\alpha = 0.82$; six items). We also probed participants' views on social inequality by asking them whether they would like to bring people above (below) them on the MacArthur socio-economic status ladder down (up) a peg or two.

- *Religious views.* We asked participants how religious they consider themselves and whether they believe that God or another spiritual non-human entity controls the events in the world[55].

## Procedure

We began data collection in the United Kingdom on 28th November 2022, with participants returning to complete the follow-up survey on 5th December 2022. We then ran a second wave of data collection in the United States on 20th June 2023, with participants returning to complete the follow-up survey on 27th June 2023. Our surveys were designed through the online survey platform Qualtrics (https://www.qualtrics.com/).

In the initial games survey, participants completed all six economic games in a random order, with punishment decisions (whether to punish a stealing target and whether to punish a target who did nothing) randomised within games. Responses to comprehension questions suggested that participants understood the six economic games (Supplementary Table 3). In order to partially mitigate the fact that some punishment strategies predict more punishment than others and are thus more 'expensive' to implement (Table 1), we chose a random game to determine bonus payments rather than summing participants' earnings across all games[62]. After all games, 62% of participants stated that they believed that their decisions had real consequences for others.

In the follow-up survey, participants completed blocks of questions on demographics, personality, Social Value Orientation, political ideology, and religious views in a random order, with questions randomised within blocks. A random decision from the Social Value Orientation Slider Measure was chosen to determine bonus payment.

Participants were paid £1.80 for completing the games survey, plus a bonus payment from the six economic games (between £0.40 and £0.70 depending on their decisions). Participants were paid £1.50 for completing the follow-up survey, plus a bonus payment from the Social Value Orientation Slider Measure (between £0.50 and £0.85 depending on their decisions).

## Statistical analysis

We pre-registered that we would use a Bayesian latent state model to infer unobserved punishment strategies from the observed data (for a similar version of this model, see ref. 63). In this model, participants $i$ in countries $c$ make binary punishment decisions across twelve decisions $j$. We assume that the probability of the observed data $y_{i,j}$ is the weighted average of the probability of the observed data conditional on each of the ten punishment strategies $s$. From this logic, the model estimates the probability of each strategy $p_s$. The full model is as follows:

$$
\begin{aligned}
y_{i,j} &\sim \text{Bernoulli}\left(\theta_j\right) \\
\theta_j &= \sum_{s=1}^{10} p_s \Pr(\text{punish}|s,j) \\
p &= \text{softmax}\left(\alpha_{c[i]}\right) \\
\alpha_{s,c} &\sim \text{Normal}(0,1)
\end{aligned}
\tag{1}
$$

The conditional probabilities $\Pr(\text{punish}|s,j)$ are hard coded in the model as outlined in Table 1. We incorporate an implementation error rate $\delta$ into these conditional probabilities by coding ticks in Table 1 with a conditional probability of $1 - \delta$ and coding crosses with a conditional probability of $0 + \delta$. We set $\delta$ to 0.05 in all models, which is similar to its value when we estimate it as a free parameter in an additional model (median posterior $\delta = 0.03$, 95% CI [0.00 0.06]; Supplementary Fig. 3). The random choice strategy is consistently coded with a conditional probability of ½ across all decisions.

To include a categorical predictor in the model, we estimate a different $\alpha_{s,c}$ for each categorical level. To include a continuous predictor $x$ in the

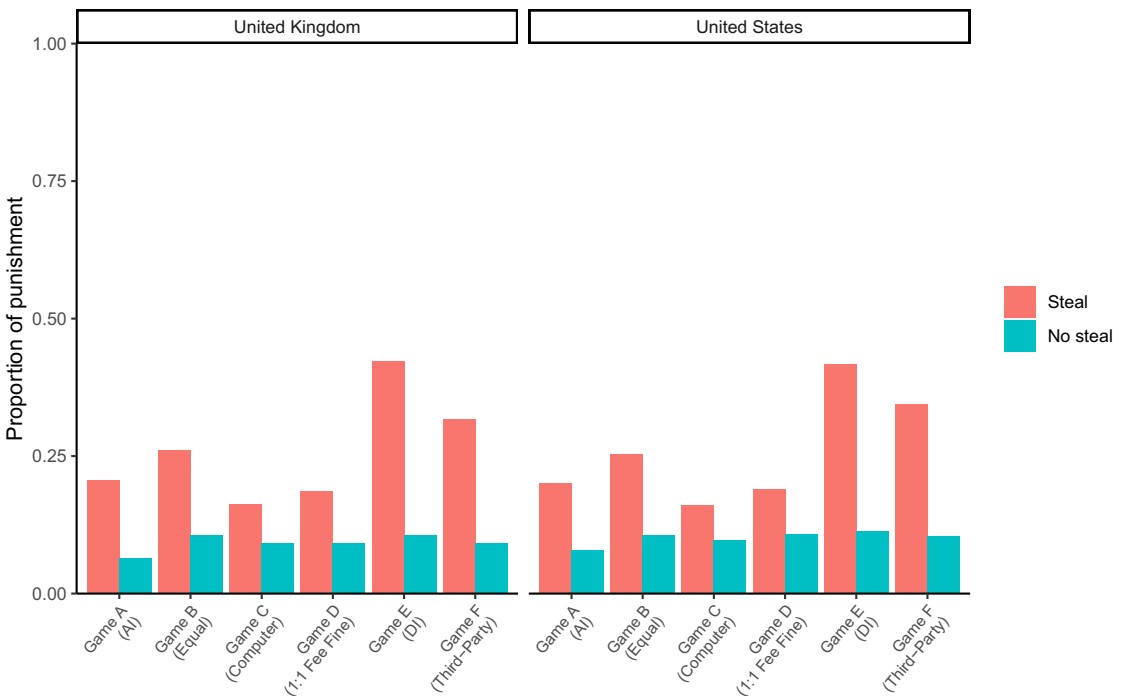

**Fig. 2 | Overall pattern of punitive behaviour across all six economic games, split by country.** $N = 2010$ participants. AI advantageous inequity, DI disadvantageous inequity.

model, we include a slope $\beta$ in the linear model for $p$:

$$y_{i,j} \sim \text{Bernoulli}\left(\theta_j\right)$$

$$\theta_j = \sum_{s=1}^{10} p_s \Pr\left(\text{punish}|s,j\right)$$

$$p = \text{softmax}\left(\alpha_{c[i]} + \beta_{c[i]}x_i\right) \quad (2)$$

$$\alpha_{s,c} \sim \text{Normal}(0, 1)$$

$$\beta_{s,c} \sim \text{Normal}(0\ 0.2)$$

These models control for multiple comparisons in several ways. First, the models estimate the effects of the predictor on all strategies simultaneously (though different predictors are necessarily included in different models). Second, the models employ strongly regularising priors on the slope parameters $\beta_{s,c}$, which makes our estimates more conservative and less susceptible to random noise[64].

We estimated the posterior distributions of these models using Hamiltonian Monte Carlo as implemented in Stan version 2.32.2[65]. We ran each model for 2000 samples, with 1000 warmup samples. R-hat values and effective sample sizes suggested that all models converged normally. Trace plots are reported in Supplementary Fig. 4.

We validated the model by simulating observed data ($n = 100$) from a known frequency of strategies. The model was successfully able to recover the known frequency of strategies from the simulated data (Supplementary Fig. 5).

### Reproducibility

All data and code are accessible on GitHub[66]: https://github.com/ScottClaessens/punishStrategies. All analyses were conducted in R version 4.4.2[67]. Visualisations were created with the *ggplot2*[68] and *cowplot*[69] R packages. We used the *targets*[70] R package and *quarto*[71] to reproducibly generate the manuscript.

### Ethical approval

This research was approved by the UCL Department of Psychology Ethics Committee (project: 3720/002) and ratified by the University of Auckland Human Participants Ethics Committee. The study was performed in accordance with all the relevant guidelines and regulations. Informed consent was obtained from all participants prior to the study, and participation was voluntary.

### Reporting summary

Further information on research design is available in the Nature Portfolio Reporting Summary linked to this article.

### Results

The overall pattern of punitive behaviour in the six economic games was very similar across both countries (Fig. 2). Participants were generally more likely to punish targets who stole compared to targets who did not steal (multilevel logistic regression; $b = 1.93$, standard error = 0.27, 95% confidence interval [1.40, 2.46], $p < 0.001$). Participants were also more likely to punish when targets' stealing behaviour generated inequalities, specifically in Games E and F ($b = 2.42$, SE = 0.44, 95% CI [1.56 3.27], $p < 0.001$).

We classified participants into a particular strategy if their behaviour across all twelve decisions matched our behavioural predictions shown in Table 1 exactly. Table 2 shows the proportion of participants following each strategy, with N/A used to represent participants who did not fit exactly into any particular strategy type. Overall, 59% of our participants could be classified exactly into one of the strategies. The most common strategy in both countries was to never punish across any of the games. The next most common strategies were those that care about minimising payoff differences (avoid disadvantageous inequity, egalitarian). Less common were the behaviour shaping strategies (deterrent, norm-enforcing), the revenge strategy, and the competitive strategies (seek advantageous inequity, competitive). Although participants often punished targets who did not steal in the six games (Fig. 2), no participants followed the antisocial strategy by exclusively punishing targets who did not steal across *all* games.

To further investigate the strategies that participants were following, we examined the most common patterns of punitive behaviour across all twelve

decisions. Supplementary Table 4 shows the proportion of participants following the 25 most common behavioural patterns, including, where appropriate, the predetermined strategies from Table 1. In both countries, a common pattern of behaviour not captured by any of the strategies was punishing only when the target stole in the third-party game (Game F). Other behavioural patterns not captured by our strategies included punishing whenever the target stole across all games and always punishing in every game, irrespective of the targets' behaviour.

While it is useful to look at exact patterns of behaviour, participants may not have implemented their chosen punishment strategy with exact precision. In reality, strategies may have been implemented probabilistically for each punishment decision. There is also the possibility of implementation errors, whereby participants occasionally 'slip up' and make decisions that are incongruent with a particular strategy. This may explain why some participants were unable to be classified exactly into a single punishment strategy.

**Table 2 | Counts and proportions of participants following each punishment strategy exactly, split by country**

| Strategy | United Kingdom (N = 1014) | | United States (N = 996) | |
|---|---|---|---|---|
| | N | Prop | N | Prop |
| Deterrent | 9 | 0.009 | 6 | 0.006 |
| Norm-enforcing | 8 | 0.008 | 16 | 0.016 |
| Revenge | 6 | 0.006 | 5 | 0.005 |
| Avoid DI | 67 | 0.066 | 62 | 0.062 |
| Egalitarian | 65 | 0.064 | 71 | 0.071 |
| Seek AI | 2 | 0.002 | 0 | 0.000 |
| Competitive | 3 | 0.003 | 1 | 0.001 |
| Antisocial | 0 | 0.000 | 0 | 0.000 |
| Never punish | 426 | 0.420 | 447 | 0.449 |
| N/A | 428 | 0.422 | 388 | 0.390 |

N/A implies that participants were unable to be classified exactly into any of the punishment strategies, AI advantageous inequity, DI disadvantageous inequity.

To deal with this complexity and include all observed data in our frequency estimates, we fitted a Bayesian latent state model to the data. This model assumes that the nine strategies in Table 1 (plus a 'random choice' strategy that chooses randomly for each decision) are the only latent strategies and that these are instantiated into observed behaviour according to the logic in Table 1 with some probability of implementation error (i.e., an intention to punish is implemented as non-punishment and vice versa). Averaging over all strategies and incorporating the possibility of implementation errors, the model estimates the probability of participants following any particular strategy, given the observed data.

The posterior estimates from the model are presented in Fig. 3. The posterior probabilities for each strategy did not differ between the two countries. In both countries, the never-punish strategy had the highest probability, followed by the egalitarian strategy. The norm-enforcing and seek advantageous inequity strategies were the next most likely, with higher posterior estimates than the competitive and antisocial strategies. None of the other strategies differed in their posterior estimates. The same general pattern emerged when we analysed the full dataset without pre-registered exclusions (Supplementary Fig. 6).

This pre-registered version of the model includes our *a priori* strategies from Table 1, but does not include other strategies that turned out to be common in the raw data. In particular, the strategy that punishes only when the target stole in the third-party game (Game F) is potentially interesting to include, as third-party punishment has been argued to reflect a uniquely-human normative psychology[37]. We therefore ran the model again, including this additional strategy (Supplementary Fig. 7). The results of this model show more tempered support for the egalitarian strategy, and increased support for both the avoid disadvantageous inequity strategy and the third-party strategy, in line with the raw sample counts. However, this result is difficult to interpret for a number of reasons. First, the behaviour of this strategy (punishing *only* in third-party contexts and *never* in second-party contexts) does not cohere with previous work showing that people who engage in third-party punishment often also engage in second-party punishment[72]. Second, while Game F is the only third-party game, it is also the game in which stealing results in the most inequality between players (a disparity of £0.40), making it difficult to know what is driving people's behaviour. For these reasons, we did not include the third-party strategy in our subsequent analyses.

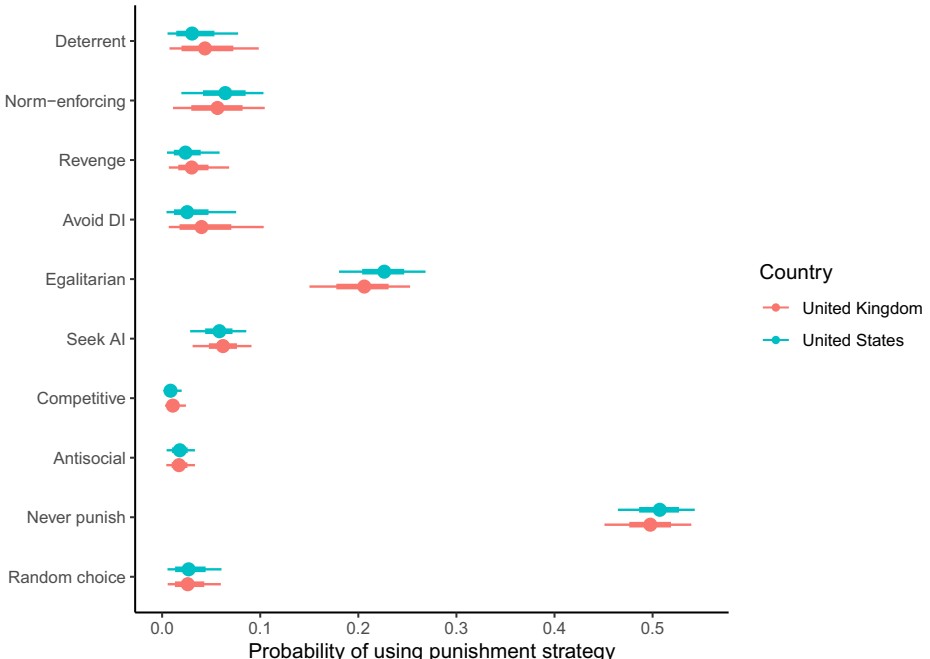

**Fig. 3 | Posterior estimates of the probabilities of following different punishment strategies from the Bayesian latent state model.** The model assumes an implementation error rate of 5%. Points represent posterior medians, line ranges represent 50% and 95% credible intervals. N = 2010 participants. AI advantageous inequity, DI disadvantageous inequity.

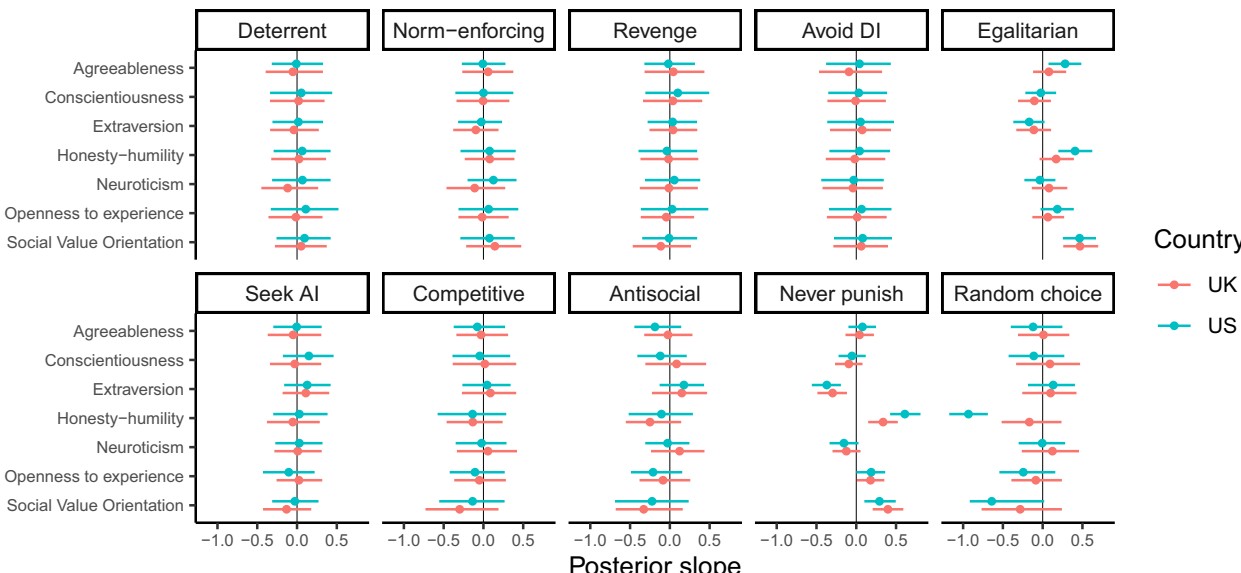

**Fig. 4 | Posterior slopes from Bayesian latent state models including Big-6 personality dimensions and Social Value Orientation.** Each row represents a separate model. All models assume an implementation error rate of 5%. Points represent posterior medians, line ranges represent 95% credible intervals. $N = 2010$ participants. AI advantageous inequity, DI disadvantageous inequity.

After estimating the relative frequencies of different strategies, we explored which traits predicted adherence to the punishment strategies. To answer this question, we included variables capturing demographics, personality, social preferences, political views, and religious views as predictors in our Bayesian latent state model. We included each variable in a separate model, predicting all ten punishment strategies (the nine from Table 1, plus the 'random choice' strategy) simultaneously.

Demographic variables tended to be unrelated to strategy usage: age and gender did not predict adherence to a particular punishment strategy (Supplementary Figs. 8 and 9). In the United States, the never-punish strategy was slightly more common among participants lower in socioeconomic status (median posterior slope = −0.20, 95% CI [−0.38 −0.03]), but this effect was small.

Conversely, personality and social preferences were linked to variation in punishment strategies. When including the Big-6 personality dimensions and Social Value Orientation (SVO) in the model, we found associations with the egalitarian, never punish, and random choice strategies, with small-to-medium effect sizes (Fig. 4). More prosocial participants (those with higher SVO scores) were more likely to follow the egalitarian and the never punish strategies, while those with lower SVO scores were more likely to enact the random choice strategy. The personality dimensions of honesty-humility and openness to experience were both positively associated with following the never punish strategy, while extraversion negatively predicted this strategy. The effects were mostly similar across countries, but occasionally differed: for example, in the United States, but not in the United Kingdom, honesty-humility was positively associated with following the egalitarian strategy and negatively associated with following the random choice strategy. Overall, the same pattern of results emerged when analysing the full dataset without exclusions (Supplementary Fig. 10).

Political and religious variables were also associated with punishment strategy (Fig. 5). These effects were small to medium in size and tended to be more pronounced in the United States. Controlling for Social Dominance Orientation, American participants higher in Right Wing Authoritarianism were more likely to follow the strategies of avoiding disadvantageous inequity and seeking advantageous inequity. Participants who stated that they would like to 'bring those below them [on the socio-economic status ladder] up a peg' were more likely to follow the egalitarian strategy, while American participants higher in Social Dominance Orientation, Right Wing Authoritarianism, and believing that God controls events in the world were

less likely to follow the egalitarian strategy. In general, religious and conservative participants were less likely to follow the never punish strategy. This general pattern of results was replicated with the full dataset (Supplementary Fig. 11).

Finally, we asked whether participants had insight into their own punishment strategy. In other words, could participants self-report the strategy that they were following during the games? To answer this question, we included participants' responses to post-game questions about their strategy as predictors in the model. As before, each predictor was included in a separate model, predicting all ten strategies simultaneously.

In general, we found that self-reported strategy usage was positively associated with the behavioural strategy that participants employed, with effects ranging from small to large in size (see Fig. 6 and Supplementary Fig. 12 for the distribution of responses to self-report questions). Figure 7 shows the relationships between self-report questions and the different punishment strategies, highlighting the combinations where the question matched the behavioural strategy. We found positive relationships between the self-report questions and strategy usage for the norm-enforcing, egalitarian, seek advantageous inequity, never punish, and random choice strategies. The 95% credible intervals for other estimates included zero, though these estimates often trended in a positive direction. The same pattern of results was found when analysing the full dataset without exclusions (Supplementary Fig. 13).

Notably, there were several instances where self-reported strategy usage predicted a *different* behavioural strategy. For example, the random choice strategy is predicted by an intention to 'make decisions at random' but also an intention to 'punish people who did not harm'. One potential explanation for this is that strategies are often motivated by multiple related reasons. Since participants reported their reasons for punishing on several slider scales, they were able to report several related motives for their behaviour. Indeed, self-reported intentions to make decisions at random and punish people who did not harm were positively correlated with one another ($\rho = 0.58$, 95% CI [0.55, 0.61], $p < 0.001$). Supplementary Fig. 14 reports the full correlation matrix for all self-reported intentions items.

## Discussion

Using a suite of economic games measuring punishment in different situations, we have shown that punishment does not serve just one function, but instead is a flexible tool that can be and is used for different purposes[6].

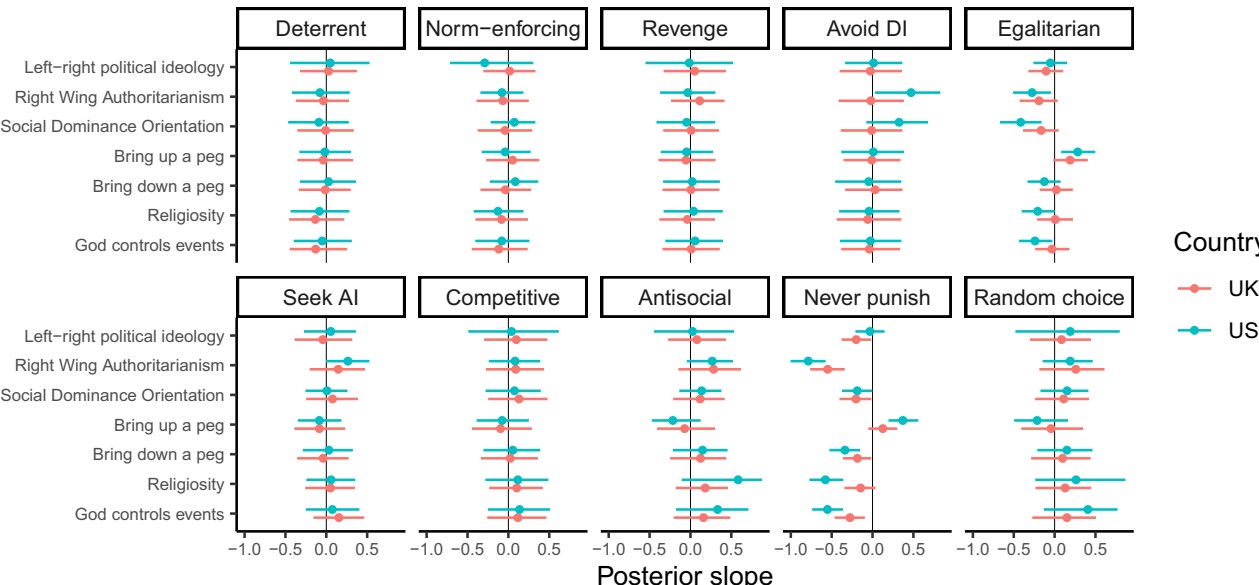

**Fig. 5 | Posterior slopes from Bayesian latent state models including political ideology, views about social inequality, and religiosity.** Each row represents a separate model aside from Social Dominance Orientation and Right Wing Authoritarianism, which control for one another within the same model. All models assume an implementation error rate of 5%. Points represent posterior medians, line ranges represent 95% credible intervals. $N = 2010$ participants. AI advantageous inequity, DI disadvantageous inequity.

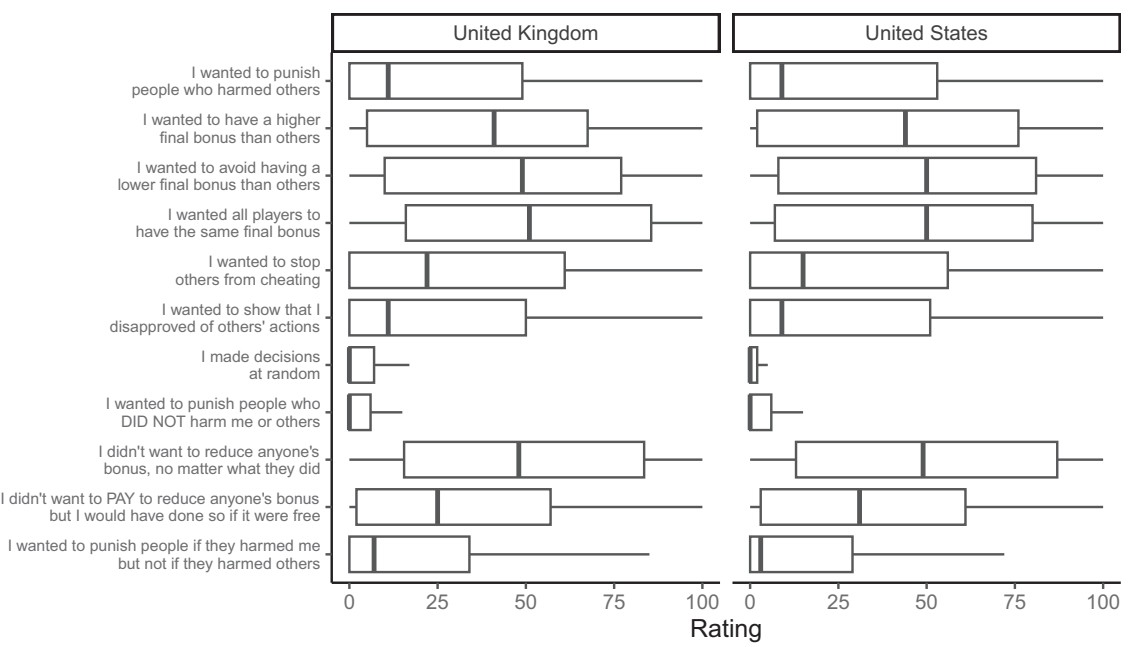

**Fig. 6 | Boxplots showing the distribution of responses to each self-report question about the reasons for participants' behaviour in the games.** $N = 2010$ participants. Boxplots represent medians and interquartile ranges.

While some use punishment to enforce norms of cooperation, others use it to reduce or even create inequality between individuals. We found that people's punishment strategy can, to some extent, be predicted by individual differences in personality, social preferences, and political and religious views. Moreover, contrary to the view that people are often unable to articulate the reasons for their punitive behaviour[30], people seem to have some degree of insight into the strategy they are using. Despite small differences, these general patterns replicated in samples from both the United Kingdom and the United States, providing further confidence in the results.

Among the punitive strategies, the most common were particularly sensitive to inequality in payoffs, either from a self-referential perspective

(i.e., avoid disadvantageous inequity) or more generally (i.e., egalitarian). This is in line with previous studies, which have highlighted inequity aversion as an important motivation for punishment in economic games[17,31,32,38,47,73]. Behaviour shaping strategies, such as deterrence and norm-enforcement, were less common than strategies sensitive to inequality in our set of games. This was reflected both in participants' elicited punishment behaviour (Fig. 3) and in their self-reports of their own strategy (Fig. 6 and Supplementary Fig. 12). Although our design did not explicitly allow for behaviour shaping as the interactions were all one-shot, we did manipulate whether the target's stealing behaviour was intentional or not (Game C), an approach which has been used in previous studies[74,75]. The

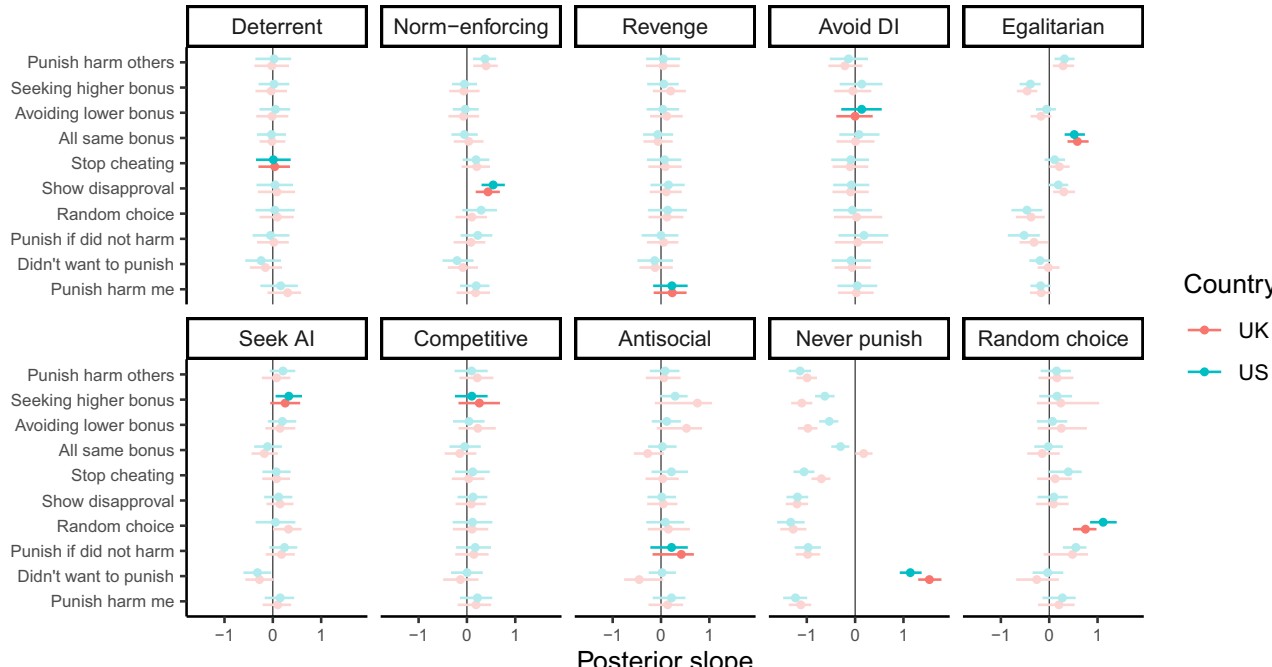

**Fig. 7 | Posterior slopes from models including self-reported strategy usage.** Each row represents a separate model. All models assume an implementation error rate of 5%. Bolded estimates represent combinations where the self-report question matched the behavioural strategy. Points represent posterior medians, line ranges represent 95% credible intervals. $N = 2010$ participants. AI advantageous inequity, DI disadvantageous inequity.

lower prevalence of behaviour shaping strategies in our study is consistent with prior work showing that punishment often continues to be used when behaviour shaping is impossible, such as when the target will never find out that they have been punished[35] or on the last round of repeated interactions[34]. We also found that participants accurately reported using the norm-enforcing strategy, but not the deterrent strategy. This finding is in line with previous research showing that people struggle to accurately report the deterrent motivations for their punitive behaviour[30].

We defined antisocial punishment as occurring when a participant exclusively punished those who did not steal. There were no participants in our sample who followed this strategy, although some players did punish non-stealing co-players. Harming non-stealing individuals was also consistent with the competitive strategies, which did appear in our sample, albeit at low frequencies. Previous work has suggested that another motive underpinning antisocial punishment is the desire to improve one's own relative payoffs[6,36]—though our data partially support this idea, if antisocial punishment was purely motivated by competitive concerns, we would also have expected to observe a reduction in antisocial punishment in Game D, where the fee-fine ratio was 1:1. That we did not suggests that the picture is more complex and that, at least in this study, antisocial punishment may not be attributable to one single motive.

The fact that people use punishment for many different reasons poses problems for the way that punishment is operationalised in classic behavioural economic game studies. In these studies, a common assumption is that participants punish to change the behaviour of cheats[11,12]. But in reality, people may be choosing the punishment option to achieve a variety of different goals. This has implications for how people respond to being targeted by punishers in these games. Targets of punishment in these studies may know that punishment reflects different motives and can respond accordingly. For example, if targets interpret punishment as serving a competitive motive, it may elicit retaliation rather than encourage cooperation[6,10,76]. As punishers' motives must be inferred (and such inferences likely depend on character traits of the target, as well as the context in which punishment occurs), there is likely to be some variation and error in attributing motives to punishers. To the extent that inferred motives affect target responses, this might help to

explain the mixed findings in the field as to whether punishment actually motivates cheating targets to subsequently cooperate[6].

It is striking that the most common strategy in our dataset was to never punish. This is partly because punishment in these games imposes an economic cost for no tangible benefit. If the fee-fine ratio had been lower, such that it was cheaper to punish, we may have seen more punishment from participants. Indeed, 72% of participants following the never punish strategy positively stated that they didn't want to pay to reduce anyone's bonus but would have done so if it were free. But the frequency of the never punish strategy perhaps also reflects a more general aversion to peer punishment, an aversion that has been highlighted in both WEIRD (Western, educated, industrialised, rich, and democratic) samples[77,78] and in small-scale societies[79]. One reason that people may be averse to peer punishment is that, because it is a fundamentally harmful act, punishment can reflect badly on the punisher, and people might therefore refrain from punishing others to avoid reputational damage[4]. People frequently avoid taking actions that could harm their reputation, even when they don't know if reputation is at stake (as in the one-shot anonymous settings used here). Another reason that people may be averse to peer punishment is that it can trigger retaliation[6]. This may be especially likely in situations that lack clear institutional norms to legitimise punishment, such as our economic games. As with reputation damage, people might abstain from peer punishment to avoid retaliation, regardless of whether retaliation is actually possible. By contrast, institutionalised punishment in small-scale societies often functions to compensate victims while limiting the potential for feuds and cycles of retaliation[80,81]. Future research should uncover whether people are more willing to punish in these conventionalised contexts (e.g., see ref. 82).

Our exploratory investigation of individual differences in punishment strategies sheds light on prior findings. For example, previous studies have found higher levels of punishment among people who are less prosocial, less agreeable, and more politically conservative[49-52]. Our results add further nuance to these previous findings. In our exploratory analyses, participants with prosocial value orientations were indeed more likely to follow the "never punish" strategy—but if they did punish, they were more likely to do so for egalitarian reasons. Agreeable participants in the United States also punished for egalitarian reasons. By contrast, religious and conservative

individuals, including individuals higher in SDO and RWA, were less likely to follow the egalitarian strategy and more likely to follow the avoid disadvantageous inequity strategy, especially in the United States. These results suggest that people with different traits may be motivated to punish others for different reasons.

## Limitations

One potential limitation with our design is that some strategies required more punishment than others, meaning that some strategies were more 'expensive' to implement. For example, the competitive strategy required punishment in ten of twelve decisions, compared to the avoid disadvantageous inequity strategy, which required only one instance of punishment (Table 1). To deal with this, we calculated participant pay-offs from a randomly-chosen game instead of accumulating the costs across all games, mitigating concerns about wealth and portfolio effects whereby participants keep track of their overall earnings over the course of the study[62]. But even accounting for this, one could still argue that the difference in overall costs explains why, for example, the competitive strategy is less common in our dataset than the avoid disadvantageous inequity strategy. We do not think this feature of our design is a concern, however, for a number of reasons. First, we are interested in measuring strategies underlying *costly* punishment. Some of these strategies, by their very nature, will be more costly to implement than others. This is reflected in our design. It would not make theoretical sense to use an alternative design where more or less punitive strategies are manipulated to cost participants the same amount. Second, when we asked participants whether they would have punished if it were free to do so, we found low agreement with this statement (Fig. 6), suggesting that participants were not particularly sensitive to the costs of punishment. In line with this, when we plot the frequencies of different strategies against their expected costs, we find that cost is not a perfect predictor of strategy frequency: many 'cheap' strategies are rare in our data and some 'expensive' strategies are quite common (e.g., always punish; Supplementary Fig. 15). Third, this argument implies that the high frequency of the never punish strategy is merely an artefact of our design, since it is the only strategy that does not cost anything to implement. But as we have discussed, many other studies have found similar aversions to costly punishment in the lab[17,32,34,43,47] and in the real world[77–79], suggesting that this result is not an artefact of our design.

Another limitation of our study is the focus on samples from the United Kingdom and the United States. Given that previous work has found substantial cross-cultural variation in both the overall prevalence of costly punishment[13] and the prevalence of particular punishment strategies, such as antisocial punishment[19], there are reasons to expect that our results might not generalise to other cultures, such as non-Western and small-scale societies. Nonetheless, it is also notable that we found striking similarities between our two samples, perhaps pointing to some cross-cultural commonalities, at least among these two WEIRD societies[83]. We also note that much previous work on punishment has focused on English-speaking Western cultures, making the strategies we identify particularly relevant to interpreting prior research.

A final potential limitation of our study is that 38% of our participants believed that their decisions in the economic games did not have real monetary consequences for others. We do not think that this is necessarily an issue for interpreting our results, since people do not tend to behave differently in incentivised vs. hypothetical economic games measuring social preferences and punitive behaviour[84]. Nonetheless, future research employing behavioural studies with online samples should monitor participants' belief in experimental paradigms, especially if incentivisation is an important feature of the design.

## Conclusion

In sum, we have shown that while many people choose not to punish peers, those who do are motivated by a variety of different concerns, including behaviour shaping, egalitarianism, and competition. Much like the observed variation in human social learning strategies[40], humans thus also exhibit variation in their punishment strategies. These individual differences map onto personality dimensions, social preferences, political and religious views, and self-reports of behaviour. We hope that future work will continue to unpack the multifaceted nature of human punishment.

## Data availability

All data used in this study are publicly available on GitHub: https://github.com/ScottClaessens/punishStrategies.

## Code availability

All code to reproduce the analyses in this study is publicly available on GitHub: https://github.com/ScottClaessens/punishStrategies.

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

## Acknowledgements

This work was supported by a Royal Society of New Zealand Catalyst Leaders Grant to Q.D.A and N.R. (ref: ILFUOA2002). The funders had no role in study design, data collection and analysis, decision to publish or preparation of the manuscript.

## Author contributions

Author roles were classified using the Contributor Role Taxonomy (CRediT; https://credit.niso.org/) as follows: Scott Claessens: conceptualization, data curation, formal analysis, investigation, methodology, visualisation, writing—original draft, and writing—review and editing. Quentin D Atkinson: conceptualization, funding acquisition, methodology, supervision, writing—original draft, and writing—review and editing. Nichola J Raihani: conceptualization, funding acquisition, investigation, methodology, supervision, writing—original draft, and writing—review and editing.

## Competing interests

The authors declare no competing interests.
