## [Transparent Peer Review file · Communications Psychology]

Individual differences in motives for costly punishment

Corresponding Author: Dr Scott Claessens

Version 0:

Decision Letter:

Dear Dr Claessens,

Thank you for your patience during the peer-review process. Your manuscript titled "Why Do People Punish? Evidence for a Range of Strategic Concerns" has now been seen by 2 reviewers, and I include their comments at the end of this message. They find your work of interest but raised some important points. We are interested in the possibility of publishing your study in Communications Psychology, but would like to consider your responses to these concerns and assess a revised manuscript before we make a final decision on publication.

We therefore invite you to revise and resubmit your manuscript, along with a point-by-point response to the reviewers. Please highlight all changes in the manuscript text file.

Editorially, we ask you to improve the presentation and discussion of the results (as raised by R1), and provide further clarification on key definitions and the justification for the hard-coded strategies in the models (as requested by R2). In addition, please conduct the additional analysis requested by R2 (i.e., re-running their Bayesian latent state models, this time hard-coding for those additional strategies that are common in the data), and address the multiple comparison concern raised by R1.

I am attaching an Editorial Requests Table that details critical reporting requirements for the revised manuscript. Please attend to each item and ensure your manuscript is fully compliant. If your revised manuscript is not aligned with these requests on major issues, such as those concerning statistics, it may be returned to you for further revisions without re-review.

Please submit the following items:

- Revised manuscript
- Point-by-point response to the referees' comments
- Cover letter (as a separate document)
- <https://www.nature.com/documents/nr-reporting-summary.pdf> Nature Research Reporting Summary
- Completed Editorial Request Table (attached).

via this link: Link Redacted .

Additional guidance is available in our style and formatting guide Communications Psychology formatting guide.

Best regards,

Yafeng Pan

Yafeng Pan, PhD
Editorial Board Member
Communications Psychology
orcid.org/0000-0002-5633-8313

REVIEWER EXPERTISE:

Reviewer #1: behavioral economics, large-scale survey

Reviewer #2: behavioral economics, large-scale survey

REVIEWER REPORTS:

Reviewer #1 (Remarks to the Author):

This is a well-crafted and carefully executed study. The methods used are sound and the results are credible and will be of interest to researchers across several disciplines. I have some suggestions to further improve the manuscript, which I list below. They mainly concern the discussion of the results presented, so I am confident that the authors will be able to address them. I strongly believe that this project has the potential to make a valuable contribution to the field.

1. While there is really a lot to like about this paper, I believe that the correlational analysis between psychological traits and punishment strategies, so the analysis and discussion around figure 4 (and to some degree figure 5) is currently its weakest part. The paper does not provide any theoretical discussion for why and how we should expect traits and punishment behavior to correlate. What kind of causality, if any, do you have in mind? An improved discussion would help the reader understand what the takeaways from this analysis are.

2. Related to this is the fact that in this section, you present many correlations, without any underlying theoretical predictions. Given multiple comparisons, how meaningful are then 95% credible intervals? How likely is it that they just represent random noise? Or to say it differently: What would we expect to find, even if all traits were unresponsive in an infinite sample? Also for this reason, it is currently hard to really get a takeaway from this section.

3. The analysis that compares participants' intentions with the algorithmic classification (figure 6) is extremely interesting, but I would recommend that the authors use these results to reflect a bit on their classification mechanism. Just to be clear, this is not at all meant as a criticism of the algorithm that the authors use. But I think it could be useful to think of what we can learn here, also for future research. For example, I note that respondents who state that they intend to punish those who did not harm, which sounds like anti-social punishment in its pure form, are often classified as following a random strategy. Is this potentially because for pro-social punishment, the classification scheme considers several fine-grained variations, but for anti-social punishment there is only one extreme variant? Or I see that the intention to punish those who harm others is positively associated with the egalitarian strategy. What could be the explanation for this?

4. The paper currently doesn't include a discussion on how generalizable the authors expect the results to be. The study was done in very particular populations (the US and the UK), and it would be interesting to learn the authors' thoughts on how universal these results are. We know that there is cultural variation in the use of punishment, see for example Hermann,

Thöni, Gächter on Anti-Social Punishment, or my own recent study on cultural variation in retaliation in the Journal of Economic Behavior & Organization (doi: 10.1016/j.jebo.2024.05.010). Should we expect then the mix of punishment strategies also to differ across cultures? And if, say, punishment is more of a cultural norm, should we expect different psychological types to engage in punishment, rather than when it is an exception?

Minor points

5. When discussing anti-social punishment, the authors write: "Instead, antisocial punishment is more likely to be motivated by improving one's relative position, which is in line with work showing that antisocial punishment disappears when relative payoffs cannot be changed (e.g., when punishment is only available with a 1:1 fee-fine ratio)"

This is a somewhat odd thing to say, given that this study finds, if anything, higher anti-social punishment in game D when the fee-fine ratio is 1:1 (figure 2). The authors should at least explain why they believe this to be the case, even if their results show the opposite.

6. In the introduction, the authors could be clearer in separating proximate and ultimate causes of punishment. For example, in evolutionary models like in references 3 or 22, it doesn't matter what the reason is why people punish, if they do it. I can also gain a reputation for being a punisher, and thus alter the way others behave towards me, even if that is not my intention.

7. When discussing the fact that some strategies are more costly to implement than others, the authors write "We employed the strategy method to deal with this, calculating participant payoffs from a randomly chosen game instead of accumulating the costs across all games. This mitigates the concern that the total cost is the most salient feature determining participants' decisions."

While I don't think that the cost differentials are a problem in the first place (strategies have different costs in real life, after all), I must admit that I don't fully understand this reasoning. In expectation, the cost differential is exactly the same when payoffs are determined with the strategy methods, rather than averaged across games. Are there any references to back up the claim that nevertheless, cost is a less salient feature when the strategy method is used? Otherwise, I would recommend simply deleting this statement.

Reviewer #2 (Remarks to the Author):

Thank you for inviting me to review this manuscript which I read with great interest. In this paper, the authors present a well-designed experiment designed to disentangle the multitude of motives that have been proposed to underlie costly punishment behavior in humans. They further address two additional questions, concerning the demographic and individual difference characteristics that relate with different punishment strategies, as well as the extent to which people can accurately describe the motives that underlie their punishment behavior. The study is well-designed and the statistical analyses are state-of-the-art. Key contributions of the paper include documenting a hesitance to punish among a large proportion of subjects, providing a clean-cut methodology for distinguishing punishment motives, and showing that some of the most prevalent motives underlying punishment are egalitarian and/or (in my reading of the results, which I will return to) norm-enforcing. The findings directly contribute to answering key questions with interdisciplinary relevance, including the prevalence of costly punishment, the functions of punishment, and the co-occurrence of prosocial and punishment motivations.

I think it is a strength of the paper that the authors made a priori decisions regarding how to define different punishment strategies based on potential underlying functions. At the same time, I think that (1) some of the decisions on definitions require further clarification and (2) that it would be worth revisiting the hard-coded strategies in their models after considering the strategies that appeared to actually be most frequent in the data.

Regarding point (1), the justification for some definitions (as described in Table 1 and the methods) was not entirely clear to me. Specifically, I was surprised that the retributive strategy was not sensitive to the intentionality of the harm imposed, whereas deterrent and norm-enforcing strategies were. Why did the authors make this choice when differentiating the above strategies? From a deterrence perspective, it might make sense to punish offenses irrespective of whether they were intended (to make sure that the message comes across that they are not tolerated). From a retributive perspective, why shouldn't one be sensitive to the intentionality of the harm imposed when deciding whether to take revenge? In the discussion section, the authors state that they "did manipulate whether the target's stealing behavior was intentional or not, an approach which has been used in previous vignette studies to identify behavior shaping motives" (see p. 20). The cited vignette studies by Carlsmith and colleagues (2002) seem to all present intentional harms and ask participants to indicate their punishment propensity, rather than manipulating intentionality as a means to identify specific kinds of motives.

Relatedly, and regarding point (2), the behavioral patterns of a large proportion of participants (42% in the UK sample and 39% in the US sample) do not fit any of the strategies pre-defined by the authors. This seems particularly high when considering that 42% of participants in the UK sample and 45% of participants in the US sample never punished. Therefore, only 16% of participants in each of the two samples could be classified into one of the predefined punishment strategies. Given that this is a relatively small proportion of each of the samples whose punishment could be classified, I think it is worth

re-visiting the main analyses after considering the strategies that actually appear common in the data. One of the most common strategies identified by the authors based on the data is a strategy that only punishes when third parties have been victimized. Arguably, this strategy is very consistent with norm enforcing motives, but is not considered as such in the main model and in the interpretation. Other strategies that are identified based on the data might similarly be consistent with norm enforcing motives, including the strategy that punishes all offenses. Therefore, my suggestion to the authors is to consider re-running their Bayesian latent state models, this time hard-coding for those additional strategies that are common in the data, and especially the strategy that only engages in third-party punishment (which seems to be the 4th most common strategy in the data, characterizing around 5% of each of the samples).

Besides these main points, I have a couple of additional suggestions:

(3) I think the authors may discuss as a limitation of the study that 48% of participants explicitly stated that they did not believe their decisions were real.

(4) The authors show the distributions of responses to the self-reported motives in the SI, Supplementary Figure 6. I think these descriptive results are interesting and could be briefly presented in the main paper. These results fit the authors' interpretation that enacted punishment is mainly motivated by considerations of relative payoffs, and especially by egalitarian motives and motives to avoid disadvantageous inequity.

Version 1:

Decision Letter:

Dear Dr Claessens,

Your manuscript titled "Why do people punish? Evidence for a range of strategic concerns" has now been seen by our reviewers, whose comments appear below. In light of their advice I am delighted to say that we are happy, in principle, to publish a suitably revised version in Communications Psychology.

We therefore invite you to revise your paper one last time to address the remaining concerns of our reviewers and a list of editorial requests. At the same time we ask that you edit your manuscript to comply with our format requirements and to maximise the accessibility and therefore the impact of your work.

EDITORIAL REQUESTS:

SUBMISSION INFORMATION:

OPEN ACCESS:

* DATA AVAILABILITY:

Link Redacted

Best regards,

Jennifer Bellingtier

Jennifer Bellingtier, PhD
Senior Editor
Communications Psychology

Yafeng Pan, PhD
Editorial Board Member
Communications Psychology
orcid.org/0000-0002-5633-8313

REVIEWER EXPERTISE:

Reviewer #1: behavioral economics, large-scale survey
Reviewer #2: behavioral economics, large-scale survey

REVIEWERS' COMMENTS:

Reviewer #1 (Remarks to the Author):

This is a careful revision that has addressed all the points I had raised. This study provides valuable insights that are both credible and of interest to a wide community.

Reviewer #2 (Remarks to the Author):

I thank the authors for taking into consideration all reviewer comments. My comments have been addressed in the revised manuscript and I have no further suggestions for improvement.

Response to Reviewers

Reviewer #1

This is a well-crafted and carefully executed study. The methods used are sound and the results are credible and will be of interest to researchers across several disciplines. I have some suggestions to further improve the manuscript, which I list below. They mainly concern the discussion of the results presented, so I am confident that the authors will be able to address them. I strongly believe that this project has the potential to make a valuable contribution to the field.

We thank the reviewer for taking the time to read through our paper and for their positive words about the study.

1. While there is really a lot to like about this paper, I believe that the correlational analysis between psychological traits and punishment strategies, so the analysis and discussion around figure 4 (and to some degree figure 5) is currently its weakest part. The paper does not provide any theoretical discussion for why and how we should expect traits and punishment behavior to correlate. What kind of causality, if any, do you have in mind? An improved discussion would help the reader understand what the takeaways from this analysis are.

We initially viewed this section of the paper as an exploratory investigation of how different punishment strategies might vary across different demographic groups and individual difference measures. In hindsight, though, we agree with the reviewer that the theoretical motivation for this section is lacking, making it difficult for the reader to extract a general takeaway from the analyses and place the findings in the context of existing literature.

To deal with this, we have expanded the Introduction section to articulate some previous findings in the literature on individual differences in punitive behaviour (lines 85-97) and have provided a general takeaway from our analyses in the Discussion section (lines 481-491). However, we do not want to engage in HARKing (hypothesising after the results are known). We have therefore clarified in the text that we did not pre-register any specific hypotheses regarding trait correlations and that the analyses are explicitly exploratory (lines 95-97).

2. Related to this is the fact that in this section, you present many correlations, without any underlying theoretical predictions. Given multiple comparisons, how meaningful are then 95% credible intervals? How likely is it that they just represent random noise? Or to say it differently: What would we expect to find, even if all traits were unpredictable in an infinite sample? Also for this reason, it is currently hard to really get a takeaway from this section.

Our Bayesian model uses strongly regularising priors on the slope parameters (a normal distribution with a mean of 0 and a variance of 0.2) which makes our estimates more

conservative and less susceptible to random noise (Gelman & Tuerlinckx, 2000). Moreover, our models predict all strategies simultaneously rather than separately, though different predictors are necessarily included in different models. These features of our analysis strategy deal with the issue of multiple comparisons. We have clarified this in our Methods section (lines 273-277).

3. The analysis that compares participants' intentions with the algorithmic classification (figure 6) is extremely interesting, but I would recommend that the authors use these results to reflect a bit on their classification mechanism. Just to be clear, this is not at all meant as a criticism of the algorithm that the authors use. But I think it could be useful to think of what we can learn here, also for future research. For example, I note that respondents who state that they intend to punish those who did not harm, which sounds like anti-social punishment in its pure form, are often classified as following a random strategy. Is this potentially because for pro-social punishment, the classification scheme considers several fine-grained variations, but for anti-social punishment there is only one extreme variant? Or I see that the intention to punish those who harm others is positively associated with the egalitarian strategy. What could be the explanation for this?

We also find it interesting that participants' self-reported reasons for punishing sometimes predicted *different* strategies in the model (Figure 7). A potential explanation for this is that strategies are often motivated by multiple related reasons. Since participants reported their reasons for punishing on several slider scales, they were able to report several related motives for their behaviour. For example, as the reviewer notes, the random choice strategy is predicted by an intention to "make decisions at random" but also an intention to "punish people who did not harm". As it happens, both of these variables are strongly positively correlated in the data ($\rho = 0.58$).

In the paper, we have included further discussion of the cases where self-reported motivations map onto different strategies (lines 401-409). As part of this discussion, we now refer to a correlation matrix of the self-report sliders in the supplement (Supplementary Figure 14). This is helpful for explaining the pattern of results in Figure 7, but may also prove useful for future research in its own right.

4. The paper currently doesn't include a discussion on how generalizable the authors expect the results to be. The study was done in very particular populations (the US and the UK), and it would be interesting to learn the authors' thoughts on how universal these results are. We know that there is cultural variation in the use of punishment, see for example Hermann, Thöni, Gächter on Anti-Social Punishment, or my own recent study on cultural variation in retaliation in the Journal of Economic Behavior & Organization (doi: 10.1016/j.jebo.2024.05.010). Should we expect then the mix of punishment strategies also to differ across cultures? And if, say, punishment is more of a cultural norm, should we expect different psychological types to engage in punishment, rather than when it is an exception?

Thank you to the reviewer for raising the important point of generalisability. We acknowledge that the two populations from which we have sampled are different to the rest of the world in many ways (Henrich *et al.*, 2010). There are reasons to expect a quite different mix of strategies in other cultures, given that prior work has found substantial cross-cultural variation in both the overall prevalence of costly punishment (Henrich *et al.*, 2006) and the prevalence of particular punishment strategies, such as antisocial punishment (Hermann *et al.*, 2008). Nonetheless, it is also notable that we found striking similarities between our UK and US samples, perhaps pointing to some cross-cultural commonalities, at least among WEIRD societies. We also note that much previous work on punishment has focused on English-speaking Western cultures, making the strategies we identify particularly relevant to interpreting prior research.

These are all fascinating questions that we hope to explore in our future work. For now, we have added a paragraph to the Discussion section which explores these points in more detail (lines 517-525).

Minor points

5. When discussing anti-social punishment, the authors write: “Instead, antisocial punishment is more likely to be motivated by improving one’s relative position, which is in line with work showing that antisocial punishment disappears when relative payoffs cannot be changed (e.g., when punishment is only available with a 1:1 fee-fine ratio)”

This is a somewhat odd thing to say, given that this study finds, if anything, higher anti-social punishment in game D when the fee-fine ratio is 1:1 (figure 2). The authors should at least explain why they believe this to be the case, even if their results show the opposite.

We agree that this statement is confusing in the light of our own data, which show that antisocial punishment does not in fact disappear with the 1:1 fee-fine ratio in Game D. We now acknowledge this in the main text and suggest that antisocial punishment may be driven by multiple motives (lines 442-447).

6. In the introduction, the authors could be clearer in separating proximate and ultimate causes of punishment. For example, in evolutionary models like in references 3 or 22, it doesn’t matter what the reason is why people punish, if they do it. I can also gain a reputation for being a punisher, and thus alter the way others behave towards me, even if that is not my intention.

We agree with the reviewer that the paper could be strengthened by distinguishing between proximate and ultimate causes of punishment. We have clarified in the Introduction section (lines 66-70) that the proximate strategies outlined in Table 1 are consistent with several underlying ultimate functions that have been studied in the evolutionary literature, including:

1. changing how others behave toward the punisher (e.g., by changing the target’s behaviour; Clutton-Brock & Parker, 1995; Raihani *et al.*, 2012)

2. changing relative payoffs (e.g., by improving the punisher's position relative to the target; Price, Cosmides & Tooby, 2002; Raihani & Bshary, 2019)
3. improving cooperation within the cultural group such that the group is more successful than other groups (e.g., by discouraging defection; Boyd *et al.*, 2003)

7. When discussing the fact that some strategies are more costly to implement than others, the authors write “We employed the strategy method to deal with this, calculating participant payoffs from a randomly chosen game instead of accumulating the costs across all games. This mitigates the concern that the total cost is the most salient feature determining participants’ decisions.”

While I don't think that the cost differentials are a problem in the first place (strategies have different costs in real life, after all), I must admit that I don't fully understand this reasoning. In expectation, the cost differential is exactly the same when payoffs are determined with the strategy methods, rather than averaged across games. Are there any references to back up the claim that nevertheless, cost is a less salient feature when the strategy method is used? Otherwise, I would recommend simply deleting this statement.

We have now clarified this discussion in a number of ways. First, we have clarified that our use of the strategy method (i.e., having participants respond to each possible partner decision) is separate from our chosen incentivisation approach (i.e., paying for a randomly-chosen decision rather than all decisions). Second, we argue that paying for a randomly-chosen decision rather than paying for all decisions mitigates concerns about wealth and portfolio effects, whereby participants keep track of their overall earnings over the course of the study (Charness *et al.*, 2016; lines 498-499). This could plausibly impact the instantiation of different strategies: for example, participants following a more 'costly' strategy may be less inclined to implement this strategy after they already have paid for several previous punishment decisions. With our “pay one” design, each punishment decision can be approached as a clean slate, even if the overall cost differential is the same.

Reviewer #2

Thank you for inviting me to review this manuscript which I read with great interest. In this paper, the authors present a well-designed experiment designed to disentangle the multitude of motives that have been proposed to underlie costly punishment behavior in humans. They further address two additional questions, concerning the demographic and individual difference characteristics that relate with different punishment strategies, as well as the extent to which people can accurately describe the motives that underlie their punishment behavior. The study is well-designed and the statistical analyses are state-of-the-art. Key contributions of the paper include documenting a hesitance to punish among a large proportion of subjects, providing a clean-cut methodology for distinguishing punishment motives, and showing that some of the most prevalent motives underlying punishment are egalitarian and/or (in my reading of the results, which I will return to) norm-enforcing. The findings directly contribute to answering key

questions with interdisciplinary relevance, including the prevalence of costly punishment, the functions of punishment, and the co-occurrence of prosocial and punishment motivations.

We thank this reviewer for their positive words about our paper, including the study design, statistical approach, and contribution to the literature.

I think it is a strength of the paper that the authors made a priori decisions regarding how to define different punishment strategies based on potential underlying functions. At the same time, I think that (1) some of the decisions on definitions require further clarification and (2) that it would be worth revisiting the hard-coded strategies in their models after considering the strategies that appeared to actually be most frequent in the data.

Regarding point (1), the justification for some definitions (as described in Table 1 and the methods) was not entirely clear to me. Specifically, I was surprised that the retributive strategy was not sensitive to the intentionality of the harm imposed, whereas deterrent and norm-enforcing strategies were. Why did the authors make this choice when differentiating the above strategies? From a deterrence perspective, it might make sense to punish offenses irrespective of whether they were intended (to make sure that the message comes across that they are not tolerated). From a retributive perspective, why shouldn't one be sensitive to the intentionality of the harm imposed when deciding whether to take revenge? In the discussion section, the authors state that they "did manipulate whether the target's stealing behavior was intentional or not, an approach which has been used in previous vignette studies to identify behavior shaping motives" (see p. 20). The cited vignette studies by Carlsmith and colleagues (2002) seem to all present intentional harms and ask participants to indicate their punishment propensity, rather than manipulating intentionality as a means to identify specific kinds of motives.

Thank you for raising this. We agree that we could have been clearer here. For deterrence, we agree that, in some contexts, punishing unintentional harms could have a deterrent effect by deterring the behaviour that caused the unintentional harm. A good example of this might be punishing a driver who causes an accident while looking at their phone or when under the influence of alcohol. In these examples, though the harm is unintentional, the act that led to the harm was *controllable*, and can therefore be 'deterred' in a way that e.g. having a heart attack and causing an accident cannot. This is sometimes called the 'controllability' effect (Cushman *et al.*, 2009). It is this logic that we are operating under with our definition of 'deterrence' – for punishment to be a deterrent, the target of punishment has to have at least some control over their actions and the consequent outcome. As Cushman *et al.* (2009) write, "it may be worthwhile to 'teach a lesson' to an accidental harm-doer... but only if she can exert at least partial control over the occurrence of future harms in similar circumstances". In our study, this was not the case: allocations in Game C were made by a computer, so punishing the unfair outcome could have no possible deterrent effect on the target.

Conversely, what we call the ‘retributive’ strategy might more accurately be labelled ‘revenge’ – it predicts punishment whenever the actor is personally harmed, even if the action was not intentional or controllable. We note that others have distinguished retribution from deterrence in that under retribution, “the punishment is an end in itself, morally justifiable regardless of its subsequent consequence for either the offender or society” (Carlsmith, 2006).

In our study, punishment of stealing in Game C (where stealing is under the computer’s control) is the only decision that distinguishes ‘retribution / revenge’ from ‘deterrence’. As argued above, we believe that for a strategy to have a deterrent motive, it must be sensitive to the intent (or controllability) of the action that should be deterred. This is not necessarily the case with retribution / revenge.

To avoid confusion, we have renamed the retribution strategy the ‘revenge’ strategy: this strategy predicts punishment in any scenario where the player was personally harmed, but is agnostic about intent and does not punish in the third-party game. We have also amended our justification for the definitions of these strategies in the text (lines 176-199).

Relatedly, and regarding point (2), the behavioral patterns of a large proportion of participants (42% in the UK sample and 39% in the US sample) do not fit any of the strategies pre-defined by the authors. This seems particularly high when considering that 42% of participants in the UK sample and 45% of participants in the US sample never punished. Therefore, only 16% of participants in each of the two samples could be classified into one of the predefined punishment strategies. Given that this is a relatively small proportion of each of the samples whose punishment could be classified, I think it is worth re-visiting the main analyses after considering the strategies that actually appear common in the data. One of the most common strategies identified by the authors based on the data is a strategy that only punishes when third parties have been victimized. Arguably, this strategy is very consistent with norm enforcing motives, but is not considered as such in the main model and in the interpretation. Other strategies that are identified based on the data might similarly be consistent with norm enforcing motives, including the strategy that punishes all offenses. Therefore, my suggestion to the authors is to consider re-running their Bayesian latent state models, this time hard-coding for those additional strategies that are common in the data, and especially the strategy that only engages in third-party punishment (which seems to be the 4th most common strategy in the data, characterizing around 5% of each of the samples).

On our calculations, the picture is slightly more optimistic: focusing only on those who punished at least once in the study, 160 out of 588 participants in the UK (27%) and 161 out of 549 participants in the US (29%) could be classified into one of our pre-registered *a priori* strategies. Nonetheless, we acknowledge the reviewer’s general point that the majority of participants were unable to be classified.

As the reviewer notes, the strategy that punishes only when the other player steals in the third-party game (Game F) is particularly common among unclassified participants in

both the UK and US. This strategy is interesting from a theoretical perspective, as third-party punishment has been argued to reflect a uniquely-human normative psychology (Fehr & Fischbacher, 2004). In response to the reviewer's request, we therefore re-ran the Bayesian latent state model hard-coding this strategy in addition to the others (Supplementary Figure 7). The results of this model show more tempered support for the egalitarian strategy, and increased support for both the avoid DI and third-party strategies, in line with the raw sample counts.

However, we have also added some discussion to the main text (lines 343-350) explaining why we think this result is difficult to interpret. First, the particular behaviour of this strategy (punishing only in third-party contexts and *never* in second-party contexts) does not cohere with previous work showing that people who engage in third-party punishment often also engage in second-party punishment (see e.g., Peysakhovich *et al.*, 2014). Second, while Game F is the only third-party game, it is also the game in which stealing results in the most inequality between players (a disparity of £0.40) making it difficult to know what is driving people's behaviour. In order to properly disentangle this strategy from others, future work would therefore need to systematically manipulate the third-party context across all game options to rule out payoff-based explanations. For these reasons, we refer to this result in the supplement, but continue to report our pre-registered analyses in the main paper.

Regarding the other strategies in the raw data (e.g., punishing stealing across all games), we have not added these to the model, as these strategies were much less common in the data, with each practiced by less than ~1% of participants. These strategies are also less theoretically meaningful and some may reflect mistakes in implementation, which our model is able to average over.

Besides these main points, I have a couple of additional suggestions:

(3) I think the authors may discuss as a limitation of the study that 48% of participants explicitly stated that they did not believe their decisions were real.

We have added this as a limitation of our study in the Discussion section (lines 526-532).

(4) The authors show the distributions of responses to the self-reported motives in the SI, Supplementary Figure 6. I think these descriptive results are interesting and could be briefly presented in the main paper. These results fit the authors' interpretation that enacted punishment is mainly motivated by considerations of relative payoffs, and especially by egalitarian motives and motives to avoid disadvantageous inequity.

We agree that these distributions are interesting. We have now moved this figure into the main paper (Figure 6).

References

- Boyd, R., Gintis, H., Bowles, S., & Richerson, P. J. (2003). The evolution of altruistic punishment. *Proceedings of the National Academy of Sciences*, *100*(6), 3531-3535.
- Carlsmith, K. M. (2006). The roles of retribution and utility in determining punishment. *Journal of Experimental Social Psychology*, *42*(4), 437-451.
- Charness, G., Gneezy, U., & Halladay, B. (2016). Experimental methods: Pay one or pay all. *Journal of Economic Behavior & Organization*, *131*, 141-150.
- Clutton-Brock, T. H., & Parker, G. A. (1995). Punishment in animal societies. *Nature*, *373*(6511), 209-216.
- Cushman, F., Dreber, A., Wang, Y., & Costa, J. (2009). Accidental outcomes guide punishment in a “trembling hand” game. *PloS One*, *4*(8), e6699.
- Fehr, E., & Fischbacher, U. (2004). Third-party punishment and social norms. *Evolution and Human Behavior*, *25*(2), 63-87.
- Gelman, A., & Tuerlinckx, F. (2000). Type S error rates for classical and Bayesian single and multiple comparison procedures. *Computational Statistics*, *15*(3), 373-390.
- Henrich, J., Heine, S. J., & Norenzayan, A. (2010). The weirdest people in the world?. *Behavioral and Brain Sciences*, *33*(2-3), 61-83.
- Henrich, J., McElreath, R., Barr, A., Ensminger, J., Barrett, C., Bolyanatz, A., ... & Ziker, J. (2006). Costly punishment across human societies. *Science*, *312*(5781), 1767-1770.
- Herrmann, B., Thoni, C., & Gächter, S. (2008). Antisocial punishment across societies. *Science*, *319*(5868), 1362-1367.
- Pedersen, E. J., McAuliffe, W. H., & McCullough, M. E. (2018). The unresponsive avenger: More evidence that disinterested third parties do not punish altruistically. *Journal of Experimental Psychology: General*, *147*(4), 514.
- Pedersen, E. J., McAuliffe, W. H., Shah, Y., Tanaka, H., Ohtsubo, Y., & McCullough, M. E. (2020). When and why do third parties punish outside of the lab? A cross-cultural recall study. *Social Psychological and Personality Science*, *11*(6), 846-853.
- Peysakhovich, A., Nowak, M. A., & Rand, D. G. (2014). Humans display a ‘cooperative phenotype’ that is domain general and temporally stable. *Nature Communications*, *5*(1), 4939.
- Price, M. E., Cosmides, L., & Tooby, J. (2002). Punitive sentiment as an anti-free rider psychological device. *Evolution and Human Behavior*, *23*(3), 203-231.
- Raihani, N. J., & Bshary, R. (2019). Punishment: one tool, many uses. *Evolutionary Human Sciences*, *1*, e12.
- Raihani, N. J., Thornton, A., & Bshary, R. (2012). Punishment and cooperation in nature. *Trends in Ecology & Evolution*, *27*(5), 288-295.